# Impact of a mass media campaign on breast cancer symptoms awareness and screening uptake in Malaysia: findings from a quasi-experimental study

Désirée Schliemann [ID],[1] Mila Nu Nu Htay,[2,3] Maznah Dahlui,[2,4] Darishiani Paramasivam,[2] Christopher R Cardwell,[1] Nor Saleha Binti Ibrahim Tamin,[5] Saunthari Somasundaram,[6] Conan Donnelly,[7] Tin Tin Su,[8] Michael Donnelly[1]

For numbered affiliations see end of article.

**Correspondence to**
Dr Désirée Schliemann;
D.Schliemann@qub.ac.uk

## ABSTRACT

**Objective** To evaluate the impact of a mass media campaign in terms of improving breast cancer (BC) symptoms awareness and screening uptake.

**Design** Before—and after—study with comparator groups.

**Setting** Selangor State, Malaysia.

**Participants** Malaysian women aged ≥40 years (n=676) from randomly selected households.

**Intervention** A culturally adapted mass media campaign (TV, radio, print media and social media).

**Primary and secondary outcome measures** The primary endpoint was BC symptoms awareness, which was assessed with the *Breast Cancer Awareness Measure* precampaign and postcampaign. Secondary outcomes included campaign reach, self-efficacy to notice BC symptoms and clinical outcomes. Clinical breast examination and mammogram screening data were collected from hospitals and clinics.

**Results** Most participants recognised at least one of the campaign materials (65.2%). The odds of seeing the campaign were lowest for Chinese women (adjusted OR 0.25, 95% CI 0.15 to 0.40) compared with Malays and for women aged ≥70 years (adjusted OR 0.47, 95% CI 0.23 to 0.94) compared with younger women. Participants who recognised the campaign were significantly more likely to have improved awareness postcampaign compared with non-recognisers particularly for key symptoms such as 'a lump or thickening in your breast' (88.9% vs 62.1%) and 'discharge or bleeding from nipple' (79.7% vs 55.3%). Improvement in symptoms awareness scores was not associated with sociodemographic variables.

**Conclusions** Implementation in Malaysia of an evidence-based mass media campaign from the UK that was culturally adapted appeared to lead to improved awareness about some BC symptoms, though various modes of media communication and perhaps other health education approaches may be required to extend the reach to diverse, multiethnic populations and all age groups.

## Strengths and limitations of this study

► The public health intervention was subjected to a systematic cultural adaptation process and drew on best available evidence.

► This was the first study to evaluate a mass media campaign designed to improve breast cancer awareness in Malaysia.

► The study outcomes were assessed using psychometrically validated measures that were pilot tested with the Malaysian multiethnic population.

► The development and evaluation approach provides a blueprint for public health researchers in other Asian countries.

► The nationwide distribution of the mass media campaign meant that it was not possible to have conventional control groups though the study was able to create internal comparator groups of 'recognisers' versus 'non-recognisers'.

common cause of cancer deaths among women in Malaysia.[2] The high cancer mortality rate is due to several factors particularly late detection. For example, between 2007 and 2011, 43% of women were diagnosed with BC at a late stage (stages 3 and 4)[1] compared with <20% of women in the UK,[3] supporting the benefits of early detection interventions that are more commonly implemented in high-income countries.[4] Late presentation is due, at least partly, to low cancer awareness. Research indicates that there is a lack of awareness among Malaysian women about common symptoms of BC,[5 6] for example, only 34% of women recognised 'a painless breast lump' as a BC symptom.[6] Other causes of delayed detection and diagnosis include denial, lack of knowledge, negative perceptions of the disease, over-reliance on traditional medicine, misperceived risk,

## INTRODUCTION

Breast cancer (BC) comprises 17.7% of all cancers in Malaysia[1] and it is the most

emotional barriers and negative perceptions towards screening.[7–9] Furthermore, current screening guidelines encourage opportunistic biennially screening for BC in Malaysian women aged between 50 and74 years[10] and population-based screening is lacking. Nationwide education programmes are encouraged as a first step to improve early detection of BC[11] in settings where population-based cancer screening is unavailable and individuals are required to self-initiate help seeking when symptoms are experienced. However, robust evaluations of mass media interventions for cancer health promotion in Asia, and particularly Malaysia, are lacking.[12] In high-income countries, mass media campaigns have improved symptoms awareness[13 14] and increased the number of BC referrals.[15] Industry and non-governmental organisations exert mass media-type efforts regularly to raise awareness about BC but these interventions tend to be short lived and are not subjected to robust evaluation.[16] In response to this knowledge gap, our collaboration developed a culturally acceptable, evidenced-based mass media campaign for Malaysia—the *Be Cancer Alert Campaign* (BCAC).[17 18] The primary endpoint of this study was BC symptoms awareness. Secondary outcomes included the reach of the campaign, perceived self-efficacy to detect symptoms, visits to a healthcare professional to discuss BC symptoms, number of BC screenings undertaken (clinical breast examination (CBE) and mammogram) and the number of BC cases diagnosed.

## METHODS

The protocol for the evaluation of the BCAC-BC was published previously[17] and is explained here in brief.

### Study population and sampling

Malaysia comprises three main ethnicities: Malay (69%), Chinese (23%) and Indian (7%).[19] The sample was drawn from Petaling District (Sungai Buloh and Petaling Jaya), Selangor State, with a multiethnic population composition.[17] Trained interviewers visited randomly selected households and invited female residents to participate if they were aged ≥40 years, spoke English or Malay, were able to provide answers without support from others and provided informed consent. Participants were interviewed 1–12 weeks before (ie, July to September 2018) and after (ie, November 2018 to January 2019) the BCAC-BC implementation.

### Intervention

Implementation of the BCAC-BC mass media campaign occurred over a 5-week period (September to October 2018). Online supplementary appendix table 1 provides a description of campaign materials. Nationwide TV and radio presented advertisements; the study area received print materials (ie, billboards, street bunting, posters and brochures); the National Cancer Society Malaysia (NCSM) delivered a social media campaign via their Facebook page. All materials contained a link to a bespoke BCAC website and highlighted the NCSM helpline.

### Patient and public involvement

This study involved the leading advocacy organisation for cancer prevention and cancer care in Malaysia—NCSM. Representatives from NCSM, the Ministry of Health (MoH), Malaysia and the university researchers worked in partnership to address the research questions. In particular, we involved our partners, members of the public and cancer survivors in focus groups and interviews regarding the preparation of the campaign materials and their delivery and we included cancer survivors in the TV advertisements. Finally, the results of the research were disseminated in collaboration with the NCSM, MoH and cancer survivors.

### Data collection

#### Questionnaire

Precampaign and postcampaign surveys were conducted to evaluate the impact of the mass media campaign on BC awareness with questions informed by the *Breast Cancer Awareness Measure*.[20] We assessed unprompted BC symptoms knowledge by asking, 'There are many warning signs and symptoms of BC. Please name as many as you can think of'. We assessed prompted knowledge by asking, 'Do you think (symptom) could be a sign for BC?' A total score for unprompted knowledge and prompted knowledge, respectively, was calculated by summing the correct answers. Sociodemographic characteristics, cancer history (of respondent) and monthly household income were assessed precampaign only. Postcampaign, specific questions assessed campaign reach.[21] Participants were asked whether they noticed the BCAC logo and other materials. The final set of questions asked participants whether or not they found the materials relevant and acceptable shared/discussed the campaign information and whether or not they or their family and/or friends visited a health professional as a result of seeing the BCAC-BC.

#### Social media monitoring

The social media activity was monitored daily by an external agency; and it was evaluated in terms of post-reach (total number of unique users who saw the advertisement/post on their Facebook feed), interaction (total number of emoji reactions including like, love, smile, wow, sad and angry), amplification (number of shares per post), conversation (number of comments per post) and total engagement (total number of interactions, amplification and conversation per post).

#### Helpline

Trained nurses recorded (with consent) NCSM helpline callers who said that they got the helpline number from one of the BCAC campaign materials in terms of date of call, gender of caller, reason for calling and campaign source for the number.

## Health service use

Staff from local health clinics and hospitals recorded the number of CBE and mammograms undertaken between July 2018 and January 2019 as well as basic sociodemographic information.

## Sample size

It was estimated that 550 participants would provide 80% power to detect, as statistically significant at a 5% level, an increase by 6% in the proportion of individuals who were aware that an unexplained lump or swelling was a symptom of BC[17] using a two-sided McNemar test.

## Data analysis

The McNemar test assessed precampaign and postcampaign proportional differences in knowledge/awareness. $\chi^2$ tests examined the associations between recognition of one or more BCAC materials when prompted (ie, 'BCAC recognisers' and 'non-recognisers') and BC awareness and tested associations between BC history or BC screening history and BC symptoms awareness. Logistic regression investigated the relationship between BCAC recognition (yes vs no) and potential explanatory variables. The final model from which adjusted estimates were calculated contained age, gender, ethnicity, marital status, education, monthly family household income, BC history and BC screening history (received BC screening—either CBE or mammogram—in the past 2 years). Similar models were applied to the outcome, 'knowledge improved' (yes vs no). Logistic regression analyses were repeated using robust standard errors to adjust for potential clustering within households[22] (the results were similar to the results that are presented here). Service utilisation data were charted over the relevant time periods. All available information was included in the analysis.

## RESULTS

The BCAC-BC was implemented as planned (except that the TV advertisement was conducted for 4 weeks instead of 5 weeks (online supplementary appendix table 1).

## Study population

A total of 992 participants completed the precampaign survey and 68% (676/992) completed the postcampaign survey (table 1). Participants who did not complete the follow-up survey could not be reached or refused to participate postcampaign. Forty-one per cent of participants who completed both surveys were aged between 40 and 49 years and about 30% of women were aged between 50 and 59 years. Malays were the most commonly represented ethnic group (51.6%), followed by Chinese (22.3%) and Indians (17.8%). Most women were married (86.8%), just over half completed secondary education (54.2%) and most (71.2%) had a monthly low household income of less than RM 4000.[23] Sixteen women (2.4%) had a personal history of BC and 25.9% underwent a mammogram in the previous 2

**Table 1** Sociodemographic characteristics of respondents precampaign and postcampaign

| | Pre only n (%) n=316 | Pre and post n (%) n=676 |
|---|---|---|
| **Age** | | |
| 40–49 years | 104 (32.9) | 274 (41.0) |
| 50–59 years | 102 (32.3) | 199 (29.8) |
| 60–69 years | 67 (21.2) | 137 (20.5) |
| ≥70 years | 41 (13.0) | 58 (8.7) |
| **Ethnicity** | | |
| Malay | 150 (47.5) | 349 (51.6) |
| Chinese | 102 (32.3) | 151 (22.3) |
| Indian | 39 (12.3) | 120 (17.8) |
| Others | 25 (7.9) | 56 (8.3) |
| **Religion** | | |
| Islam | 169 (53.5) | 406 (60.1) |
| Christianity | 26 (8.2) | 50 (7.4) |
| Buddhism | 81 (25.6) | 109 (16.1) |
| Hinduism | 31 (9.8) | 100 (14.8) |
| Others | 3 (0.9) | 6 (0.8) |
| **Marital status** | | |
| Married | 265 (83.9) | 587 (86.8) |
| Single | 51 (16.1) | 89 (13.2) |
| **Education** | | |
| No formal education | 45 (14.2) | 88 (13.0) |
| Primary | 52 (16.5) | 97 (14.4) |
| Secondary | 158 (50.0) | 366 (54.2) |
| Tertiary | 59 (18.7) | 124 (18.4) |
| **Family income** | | |
| <RM4000 | 189 (59.8) | 457 (71.2) |
| RM4000–10 000 | 72 (22.8) | 140 (21.8) |
| >RM10 000 | 23 (7.3) | 45 (7.0) |
| **BC history (self only)** | | |
| No | 307 (97.2) | 660 (97.6) |
| Yes | 9 (2.8) | 16 (2.4) |
| **BC screening history*** | | |
| No | 222 (70.3) | 498 (75.5) |
| Yes | 94 (29.7) | 175 (25.9) |

Missing information: age (n=3), religion (n=5), monthly family income (n=34), BC screening history (n=3).
*BC screening history refers to mammogram in the past 2 years.
†Participants who are widowed, divorced and who never married.
‡No formal education—includes never schooled/ never completed primary school; primary education—includes completed primary school; secondary education—includes completed form 3/completed form 5/certificate/A-level/ Malaysian Higher School Certificate (STPM)/ Higher School Certificate (HSC); tertiary education— includes diploma/ bachelor degree/ postgraduate degree.
§Income of all household member combined.
BC, breast cancer.

years. Women who completed the follow-up assessment were significantly younger compared with women who completed only the precampaign survey. There was a

higher proportion of Malay and Indian participants and fewer Chinese participants at follow-up compared with baseline (table 1).

## Campaign reach

Eighteen per cent of participants reported that they previously saw the BCAC logo. Participants remembered (unprompted) seeing BCAC posters in clinics (11.7%), TV advertisements (9.3%), outdoor materials (bunting/billboards, 7.4%), newspaper articles (4.3%) and hearing BCAC radio advertisements (2.4%) (online supplementary appendix figure 1). Furthermore, 26.9% recognised the slogan, '*Don't be shy to check your breast*'. When participants were prompted or shown BCAC materials, 65.2% recognised BCAC materials (ie, 47.9% recognised the TV advertisements, 29% recognised print materials and 22.6% recognised the radio advertisement). TV advertisements were deemed thought-provoking by 60% and relevant by 69% (online supplementary appendix figure 2). Print materials were thought-provoking for 40% and relevant for 51.3%; fewer participants described radio ads in these terms (19.4% and 25.9%). Almost 40% discussed the advertisements with friends/family and 21.7% stated that they or their family/friends went to see a doctor as a result of seeing the advertisements. TV advertisements were most commonly recognised by Malays (66.4%) followed by 'others' (45.5%). Indian participants (61.3%) recognised radio advertisements more often than other ethnic groups. Print advertisements were recognised by more Malays (37%) compared with between 19.5% and 25.5% of other ethnicities (online supplementary appendix figure 3). Between-ethnic group differences in campaign reach were confirmed by regression analysis (table 2). The odds that participants saw one or more of the BCAC materials (when prompted) were significantly lower for Chinese and 'others' compared with Malays (adjusted OR 0.25, 95% CI 0.15 to 0.40, p<0.001 and OR 0.34, 95% CI 0.18 to 0.65, p=0.001). Campaign reach towards people aged ≥70 years appeared to be relatively poor (adjusted OR 0.47, 95% CI 0.23 to 0.94, p=0.032).

Thirty-two social media 'posts' were shared through Facebook and 11/32 posts were boosted. The boosted posts reached significantly more people than 'ordinary' posts. The boosted post describing BC symptoms in Malay achieved the highest reach (reach: 202 430; total engagement: 4498 and shares: 1379) compared with the same post in English (reach: 18 012, total engagement: 126 and shares: 25). The second most popular post showed highlights of the campaign launch (reach: 19 071, total engagement: 816 and shares: 45) and a post on breast self-examination in English (reach: 15 673, total engagement: 322 and shares: 87). Posts that were not boosted reached between 1495 and 486 Facebook users, whereas posts that were boosted reached between 202 430 and 3412 users.

The helpline received five calls from women (aged 22–42 years) who called after seeing a BCAC advertisement; 4/5 women reported experiencing potential BC symptoms and one woman had had a negative biopsy result and enquired about follow-up appointments.

## Campaign impact

Knowledge improved significantly for six BC symptoms (unprompted): 'pain in one of your breasts or armpits' (18.9% vs 23.2%), 'discharge or bleeding from nipple' (10.1% vs 16.4%), 'nipple rash' (1.3% vs 3.6%), 'redness of your breast skin' (1.5% vs 7.2%), 'lump or thickening under armpit' (3.4% vs 7.5%) and 'changes in the shape of your breast or nipple' (1.5% vs 3.7%) significantly at follow-up (table 3). Knowledge/awareness (prompted) increased significantly for three BC symptoms: 'change in the position of your nipple' (58.7% vs 67.3%), 'pain in one of your breasts or armpits' (72.5% vs 77.5%) and 'redness of your breast skin' (54.9% vs 62.4%). Overall symptom knowledge/awareness scores (prompted) improved significantly at follow-up (premean 7.45 (SD: 3.05) and postmean 7.84 (SD: 2.86)). BCAC recognisers identified on average a higher number of BC symptoms at follow-up compared with baseline (premean 7.68 (SD: 2.89) and postmean 8.26 (SD: 2.46)). There was some evidence that the improvement in awareness by BCAC recognisers was higher compared with non-recognisers (mean change 0.59 (SD: 3.29) vs mean change −0.24 (SD: 3.88)) (online supplementary appendix table 2).

A significantly higher proportion of BCAC recognisers compared with non-recognisers who were not aware of BC symptoms at baseline improved their knowledge (unprompted) about the symptom 'a lump or thickening in your breast' at follow-up (55.9% vs 41.0%) (table 3). Furthermore, significantly more BCAC recognisers improved their awareness (prompted) about the following symptoms 'change in the position of your nipple' (62.7% vs 41.6%), 'discharge or bleeding from nipple' (79.7% vs 55.3%), 'a lump or thickening in your breast' (88.9% vs 62.1%) and 'changes in the size of your breast or nipple' (67.2% vs 46.2%) compared with non-recognisers. The proportion of women who were very/fairly confident that they would recognise a BC symptom increased at follow-up (58.9% vs 68.9%) (table 3) and there was no significant difference between BCAC recognisers and non-recognisers (63.7% vs 54.4%). Precampaign, 96.1% of women reported that they would see their doctor within 2 weeks if they noticed a BC symptom and there was no change at follow-up. Regression analysis indicated that none of the sociodemographic or campaign recognition variables exerted a marked influence on BC symptoms awareness postcampaign after adjustment (table 4).

From July 2018 to January 2019, 29 000 CBEs (figure 1) as well as 2051 mammograms were performed (figure 2). More mammograms and CBEs were conducted in October (337 and 4792), January (335 and 4978) and July (331 and 4415) compared with other months. Most CBEs were performed on Malay ethnic women (85%) (online

**Table 2** The relationship between the sociodemographic characteristics of respondents and their recognition of any aspect of the BCAC-BC campaign*

| | n (%) | OR (95% CI) (unadjusted) | P | OR (95% CI) (adjusted)† | P |
|---|---|---|---|---|---|
| **Age** | | | | | |
| 40–49 years | 198/274 (72.3) | Reference | | Reference | |
| 50–59 years | 132/199 (66.3) | 0.72 (0.48 to 1.09) | 0.120 | 0.74 (0.47 to 1.16) | 0.187 |
| 60–69 years | 78/137 (56.9) | 0.48 (0.31 to 0.75) | 0.001 | 0.65 (0.39 to 1.10) | 0.107 |
| ≥70 years | 28/58 (48.3) | 0.33 (0.18 to 0.60) | <0.001 | 0.47 (0.23 to 0.94) | 0.032 |
| **Ethnicity** | | | | | |
| Malay | 265/349 (75.9) | Reference | | Reference | |
| Chinese | 68/151 (45.0) | 0.25 (0.16 to 0.37) | <0.001 | 0.25 (0.15 to 0.40) | <0.001 |
| Indian | 81/120 (67.5) | 0.62 (0.39 to 0.99) | 0.046 | 0.69 (0.42 to 1.14) | 0.145 |
| Others | 27/56 (48.2) | 0.29 (0.16 to 0.52) | <0.001 | 0.34 (0.18 to 0.65) | 0.001 |
| **Marital status** | | | | | |
| Married | 380/566 (67.1) | Reference | | Reference | |
| Single | 61/86 (70.9) | 1.19 (0.73 to 1.96) | 0.484 | 1.26 (0.72 to 2.20) | 0.422 |
| **Education** | | | | | |
| No formal education | 46/88 (52.3) | Reference | | Reference | |
| Primary | 57/97 (58.8) | 1.34 (0.74 to 2.44) | 0.333 | 1.33 (0.68 to 2.57) | 0.404 |
| Secondary | 255/366 (69.7) | 2.14 (1.32 to 3.48) | 0.002 | 1.65 (0.93 to 2.91) | 0.085 |
| Tertiary | 82/124 (66.1) | 1.99 (1.11 to 3.55) | 0.021 | 1.98 (0.93 to 4.18) | 0.075 |
| **Monthly family income** | | | | | |
| <RM4000 | 311/443 (70.2) | Reference | | Reference | |
| RM4000–RM10 000 | 82/135 (60.7) | 0.66 (0.44 to 0.98) | 0.040 | 0.73 (0.46 to 1.18) | 0.202 |
| >RM10 000 | 27/43 (62.8) | 0.72 (0.37 to 1.37) | 0.315 | 0.88 (0.41 to 1.86) | 0.728 |
| **BC history** | | | | | |
| No | 430/660 (65.2) | Reference | | Reference | |
| Yes | 11/16 (68.8) | 1.05 (0.36 to 3.07) | 0.923 | 1.21 (0.37 to 3.98) | 0.754 |
| **BC screening history‡** | | | | | |
| No | 337/510 (66.1) | Reference | | Reference | |
| Yes | 103/163 (63.2) | 0.93 (0.63 to 1.36) | 0.705 | 1.03 (0.67 to 1.60) | 0.893 |

n—number of participants 'reached' or who reported that they saw (one or more parts of) the campaign divided by the total number of survey participants.
*This includes participants who reported that they have been exposed to either the BCAC-BC TV, radio and/or print materials when prompted with the advertisement at follow-up.
†Adjusted for age, ethnicity, marital status, education, monthly family income, BC history and BC screening history.
‡If people replied 'yes' to BC history and BC screening history, they were coded as 'yes' for BC history and 'no' for BC screening history in this model.
BC, breast cancer; BCAC, Be Cancer Alert Campaign.

supplementary appendix table 3) and women aged 20–29 years (58%) and 30–39 years (37%). Mammograms were undertaken for Malays (56%), Chinese (27%) and Indians (16%).

## DISCUSSION

Research-informed BC mass media campaigns with robust evaluation are lacking in Asian countries like Malaysia.[16] The need to improve BC awareness was confirmed by the limited symptoms awareness and low confidence to notice symptoms that were identified in the precampaign survey results. The majority of women noticed the BCAC-BC materials, in particular, the TV advertisements (even though numerous BC awareness activities took place in October as part of BC awareness month); and they found the culturally adapted campaign materials acceptable and relevant to their life circumstances. The salience of TV advertisements as the most commonly recognised campaign material is in keeping with other robust evaluations.[13 24] Furthermore, awareness about

**Table 3** Be Cancer Alert Campaign—breast cancer awareness precampaign and postcampaign (n=676)

| Survey question | Pre n (%) | Post n (%) | P (McNemar) | Knowledge improvement in BCAC recognisers* n (%) | Knowledge improvement in BCAC non-recognisers† n (%) | P ($\chi^2$) |
|---|---|---|---|---|---|---|
| Signs and symptoms (unprompted) | | | | | | |
| Change in the position of your nipple | 9 (1.3) | 7 (1.0) | 0.804 | 7/435 (1.6) | 0/208 (0.0) | 0.152 |
| Pulling in your nipple | 7 (1.0) | 16 (2.4) | 0.064 | 11/438 (2.5) | 3/207 (1.4) | 0.565 |
| Pain in one of your breasts or armpits | 128 (18.9) | 157 (23.2) | 0.041 | 74/351 (21.1) | 27/179 (15.1) | 0.122 |
| Puckering or dimpling of your breast skin | 4 (0.6) | 7 (1.0) | 0.549 | 5/439 (1.1) | 2/210 (1.0) | 0.999 |
| Discharge or bleeding from your nipple | 68 (10.1) | 111 (16.4) | <0.001 | 55/401 (13.7) | 19/188 (10.1) | 0.272 |
| A lump or thickening in your breast | 440 (65.1) | 437 (64.6) | 0.897 | 80/143 (55.9) | 34/83 (41.0) | 0.042 |
| Nipple rash | 9 (1.3) | 24 (3.6) | 0.012 | 19/437 (4.3) | 4/207 (1.9) | 0.188 |
| Redness of your breast skin | 10 (1.5) | 49 (7.2) | <0.001 | 35/436 (8.0) | 11/207 (5.3) | 0.279 |
| Lump or thickening under your armpit | 23 (3.4) | 51 (7.5) | 0.001 | 35/426 (8.2) | 10/205 (4.9) | 0.174 |
| Changes in the size of your breast or nipple | 12 (1.8) | 18 (2.7) | 0.327 | 13/435 (3.0) | 2/206 (9.7) | 0.194 |
| Changes in the shape of your breast or nipple | 10 (1.5) | 25 (3.7) | 0.015 | 16/434 (3.7) | 8/208 (3.8) | 1.000 |
| Signs and symptoms (prompted) | | | | | | |
| Change in the position of your nipple | 397 (58.7) | 455 (67.3) | <0.001 | 106/169 (62.7) | 42/101 (41.6) | 0.001 |
| Pulling in your nipple | 391 (57.8) | 397 (58.7) | 0.745 | 85/182 (46.7) | 32/92 (34.8) | 0.079 |
| Pain in one of your breasts or armpits | 490 (72.5) | 524 (77.5) | 0.029 | 81/110 (44.5) | 45/69 (65.2) | 0.302 |
| Puckering or dimpling of your breast skin | 379 (56.1) | 380 (56.2) | 0.999 | 100/187 (53.5) | 43/97 (44.3) | 0.181 |
| Discharge or bleeding from your nipple | 557 (82.4) | 560 (82.8) | 0.814 | 55/69 (79.7) | 26/47 (55.3) | 0.009 |
| A lump or thickening in your breast | 598 (88.5) | 612 (90.5) | 0.211 | 40/45 (88.9) | 18/29 (62.1) | 0.014 |
| Nipple rash | 345 (51.0) | 367 (54.3) | 0.227 | 111/213 (52.1) | 46/109 (42.2) | 0.117 |
| Redness of your breast skin | 371 (54.9) | 422 (62.4) | 0.003 | 112/193 (58.0) | 47/103 (45.6) | 0.055 |

Table 3   Continued

| Survey question | Pre n (%) | Post n (%) | P (McNemar) | Knowledge improvement in BCAC recognisers* n (%) | Knowledge improvement in BCAC non-recognisers† n (%) | P ($\chi^2$) |
|---|---|---|---|---|---|---|
| Lump or thickening under your armpit | 549 (81.2) | 566 (83.7) | 0.213 | 54/68 (79.4) | 34/54 (63.0) | 0.070 |
| Changes in the size of your breast or nipple | 476 (70.4) | 497 (73.5) | 0.173 | 78/116 (67.2) | 36/78 (46.2) | 0.005 |
| Changes in the shape of your breast or nipple | 486 (71.9) | 511 (75.6) | 0.105 | 80/114 (70.2) | 38/69 (55.1) | 0.056 |
| | | | | Improved attitudes in recognisers* | Improved attitudes in non-recognisers† | |
| How confident are you that you would notice a BC symptom? (*those very confident/fairly confident*) | 391 (58.9) | 439 (68.9) | <0.001 | 100/157 (63.7) | 56/103 (54.4) | 0.170 |
| How soon would you go and see a doctor if you noticed a BC sign/symptom? (*those within 2 weeks*) | 614 (96.1) | 637 (97.1) | 0.337 | 19/20 (95.0) | 13/14 (92.9) | 0.999 |

Missing information: confidence pre n=12; confidence post n=39; delayed help seeking pre n=20; delayed help seeking post n=37.
*This includes participants who reported that they have been exposed to either the BCAC-BC TV, radio and/or print materials when prompted at follow-up.
†This includes all participants who reported that they have not seen any of the BCAC-BC TV, radio and/ or print materials when prompted at follow-up.
BC, breast cancer; BCAC, Be Cancer Alert Campaign.

individual BC symptoms was higher among BCAC recognisers compared with non-recognisers. The high reach and improvement in symptoms awareness suggested that the campaign had an impact in terms of improving awareness about some BC symptoms.

Campaign reach varied between ethnicities, though improved awareness of BC symptoms did not appear to vary across sociodemographic variables including ethnicity. The differential reach that was experienced by ethnic groups was similar to the findings from the BCAC for colorectal cancer (BCAC-CRC) in Malaysia.[25] Different attitudes and behaviours across ethnic groups regarding help seeking may help to explain the apparent discrepancy. For example, US health survey participants who did not seek health information in the media and preferred to trust their health service provider tended to be older and have a lower socioeconomic status.[26] Mass media as a public health programme or intervention requires careful tailoring to maximise the reach and impact of given public health messages.

Although the BCAC-BC had greater reach compared with the BCAC-CRC, the BCAC-CRC appeared to be more successful in terms of improving symptoms awareness[25], perhaps, because of the higher precampaign symptom knowledge/awareness (prompted and unprompted) among the BCAC-BC sample compared with BCAC-CRC participants. Awareness improved significantly at follow-up for three prompted BC symptoms, whereas awareness about all prompted CRC symptoms was improved. The success of the BCAC-BC was due partly to the systematic cultural adaptation process[18] that was undertaken in order to address, for example, the need to refer to breast-related issues with contextual sensitivity and to include only information about a lump. Although a lump appeared already to be a commonly known symptom, it was unacceptable culturally to present other symptoms of BC. Such restrictions did not apply to the BCAC-CRC.

BC is the best-known cancer site in most countries. It was the most common cancer reported in major online

Table 4  Improvement in overall prompted symptom awareness by sociodemographic characteristics and recognition of BCAC-BC advertisements (binary logistic regression)

| | n (%) | OR (95% CI) (unadjusted) | P | OR (95% CI) (adjusted)* | P |
|---|---|---|---|---|---|
| **Age** | | | | | |
| 40–49 years | 122/274 (44.5) | Reference | | Reference | |
| 50–59 years | 101/199 (50.8) | 1.28 (0.89 to 1.85) | 0.181 | 1.36 (0.91 to 2.04) | 0.130 |
| 60–69 years | 60/137 (43.8) | 0.97 (0.64 to 1.47) | 0.888 | 0.97 (0.60 to 1.56) | 0.890 |
| ≥70 years | 26/58 (44.8) | 1.01 (0.57 to 1.79) | 0.966 | 0.99 (0.51 to 1.93) | 0.982 |
| **Ethnicity** | | | | | |
| Malay | 162/349 (46.4) | Reference | | Reference | |
| Chinese | 63/151 (41.7) | 0.83 (0.56 to 1.22) | 0.333 | 0.82 (0.51 to 1.33) | 0.420 |
| Indian | 58/120 (48.3) | 1.08 (0.71 to 1.64) | 0.717 | 1.26 (0.75 to 2.12) | 0.392 |
| Others | 30/56 (53.6) | 1.33 (0.76 to 2.35) | 0.321 | 1.27 (0.67 to 2.41) | 0.461 |
| **Marital status** | | | | | |
| Married | 270/587 (46.0) | Reference | | Reference | |
| Single | 43/89 (48.3) | 1.10 (0.70 to 1.72) | 0.683 | 1.17 (0.72 to 1.90) | 0.532 |
| **Education** | | | | | |
| No formal education | 39/88 (44.3) | Reference | | Reference | |
| Primary | 56/97 (57.7) | 1.72 (0.96 to 3.07) | 0.069 | 1.50 (0.80 to 2.81) | 0.210 |
| Secondary | 156/366 (42.6) | 0.93 (0.58 to 1.49) | 0.773 | 0.93 (0.54 to 1.59) | 0.789 |
| Tertiary | 62/124 (50.0) | 1.26 (0.73 to 2.17) | 0.415 | 1.53 (0.77 to 3.04) | 0.227 |
| **Monthly family income** | | | | | |
| <RM4000 | 210/457 (46.0) | Reference | | Reference | |
| RM4000–RM10 000 | 67/140 (47.9) | 1.08 (0.74 to 1.58) | 0.693 | 1.15 (0.74 to 1.79) | 0.528 |
| >RM10 000 | 19/45 (42.2) | 0.86 (0.46 to 1.60) | 0.632 | 0.84 (0.42 to 1.71) | 0.636 |
| **BC history** | | | | | |
| No | 307/660 (46.5) | Reference | | Reference | |
| Yes | 6/16 (37.5) | 0.69 (0.25 to 1.92) | 0.477 | 0.54 (0.18 to 1.65) | 0.281 |
| **BC screening history** | | | | | |
| No | 237/510 (46.5) | Reference | | Reference | |
| Yes | 75/163 (46.0) | 0.98 (0.69 to 1.40) | 0.919 | 0.83 (0.56 to 1.24) | 0.358 |
| **TV ad recognition** | | | | | |
| No | 155/347 (44.7) | Reference | | Reference | |
| Yes | 154/324 (47.5) | 1.12 (0.83 to 1.52) | 0.457 | 1.22 (0.85 to 1.77) | 0.281 |
| **Radio ad recognition** | | | | | |
| No | 240/511 (47.0) | Reference | | Reference | |
| Yes | 68/153 (44.4) | 0.90 (0.63 to 1.30) | 0.583 | 0.79 (0.51 to 1.22) | 0.289 |
| **Print ad recognition** | | | | | |
| No | 221/468 (47.2) | Reference | | Reference | |
| Yes | 88/196 (44.9) | 0.91 (0.65 to 1.27) | 0.584 | 0.96 (0.66 to 1.40) | 0.847 |

*Adjusted for age, ethnicity, marital status, education, monthly family income, BC history, BC screening history, TV ad recognition, radio ad recognition, print ad recognition.
BC, breast cancer; BCAC, Be Cancer Alert Campaign.

newspapers in the USA[27] and it is the most 'promoted' cancer in Malaysia.[16] Also, online information-seeking behaviour related to BC is higher during October (BC awareness month).[28 29] Findings from the BCAC social media campaigns demonstrated higher reach and engagement for the BC social media posts compared with the CRC posts (ie, highest reach of CRC-related post: 92 678 users vs BC-related post: 202 430 users;

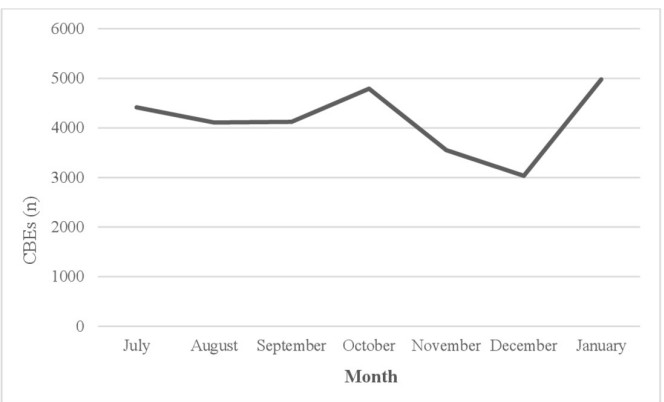

**Figure 1** Clinical breast examinations (CBEs) undertaken in four public clinics (Petaling District) between July 2018 and January 2019.

highest engagement with CRC-related post: 2065 users vs BC-related post: 4498 users).[25] CRC campaign material about symptoms may be perceived to be less pleasant to share compared with BC symptoms. Overall, it may not be surprising that BC awareness levels are higher than other cancer sites such as CRC and that the gains from a mass media campaign directed towards BC would be smaller given this elevated baseline awareness. It might be more impactful to reserve the use of mass media campaigns for areas where there is a known low level of awareness and knowledge. The number of calls reported by NCSM as a result of both BCAC campaigns was low compared with caller volume reported in other campaign evaluations, for example, a print and online media campaign targeting low-income African American women for 2.5 months received 97 calls.[30]

The apparently high number of CBEs was encouraging and may suggest good implementation of CBE guidelines among younger women (ie, CBE every 3 years) but not regarding women aged ≥40 years (5.4%) (ie, an annual CBE).[31] Previous research on CBE uptake in Malaysia found similar CBE rates across age groups.[32] The fact that data reflect public health clinic activity may explain why most CBEs were performed on Malay women (85%)—that is, public health clinics were used mainly by

Malays while Chinese tended to visit private clinics for screening activity though they may avail of public healthcare depending on costs). Mammogram activity in this study did not appear to be affected by ethnicity and was conducted among women aged ≥40 years (98.4%), which is in line with BC screening guidelines (ie, mammograms biennially for women aged 50–74 years).[10] Yet, the relatively low number of mammograms may highlight gaps in secondary BC prevention and care in Malaysia. Health service uses findings from the *Be Clear on Cancer* campaign in England for women aged 70 years and older suggested that though screening referrals increased, the number of cancer cases detected did not increase, thereby raising questions about the value of the campaign in the face of the higher workload and low conversion rate.[15] Findings from studies in the USA present mixed results about improved screening uptake. For example, one study found that screening activity in November (following BC awareness month) increased during the 90s when BC awareness and screening were low; however, increments in screening activity of this kind lessened with time as overall screening levels reached a high threshold.[33] The USA Health Communication Survey found that TV viewing (news or entertainment) prompted health information-seeking behaviour beyond sociodemographic variables and influenced cancer screening behaviour.[34] Data covered only 7 months and CBE data and private clinic data were unavailable. Mindful of service data limitations is the first study to present CBE and mammogram activity over a series of consecutive months in Malaysia. The highest number of mammograms and CBEs was conducted in July, October and January. October is 'breast cancer awareness month' and June and December are holiday months. Although a public awareness BC campaign may increase screening rates, due to widespread routine screening activity throughout the year, campaigns may lose impact regarding the detection of BC cases (ie, people at low-risk or medium-risk and the 'worried well' may be more likely to respond to cancer awareness campaigns).[33 35] BC campaign activities other than those related to the BCAC-BC may also have influenced participants' responses; and this potential limitation needs to be kept in mind when interpreting findings regarding the improvement of BC awareness. In addition, participants may have improved their knowledge due to the precampaign survey though the results from the comparative analysis of BCAC recognisers and nonrecognisers who did not support this interpretation.

A recent report[36] stated that evidence about the role and effectiveness of media channels in public health campaigns is limited due, partly and for obvious reasons, to lack of controlled studies and randomised control trials. There have been very few mass media campaigns in Asia that have been scientifically evaluated.[12] This study afforded a degree of control by creating and comparing internal groups based on whether or not they recognised the campaign. It is a reasonable argument, that longer and more intensive campaigns would be more impactful[36 37].

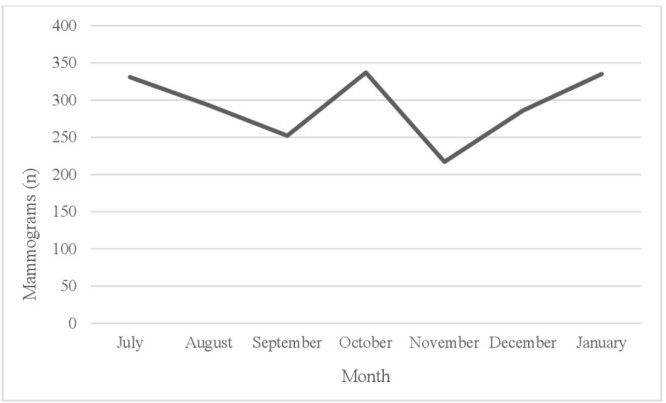

**Figure 2** Mammograms undertaken in two public hospitals (Petaling District) between July 2018 and January 2019.

However, our evaluation demonstrated that even with a modest and short-term campaign, it is possible to improve knowledge about BC symptoms and influence help-seeking behaviour. Of course, the long-term impact of the campaign and sustainability of the improvements are unknown.

## CONCLUSION

The BCAC-BC study addresses a knowledge gap and adds important insights into the impact of mass media for BC health promotion as well as implementation challenges related to (Southeast) Asia. Overall, the findings presented here favour positively the use of mass media in breast health promotion. Future campaigns may be enhanced via increased collaboration with disciplines such as social marketers and health journalism.[38] There appears to be a need for mass media campaigns to be tailored to particular subpopulations or hard-to-reach subgroups, especially in the context of multicultural societies. Women aged ≥70 years, for example, were less likely to observe or notice the BCAC-BC materials and to have received a mammogram (5%). Targeting older women and women from ethnic minorities may be an important consideration in efforts to improve the reach of future BC awareness activities in Malaysia.[15]

**Author affiliations**
[1]Centre for Public Health and UKCRC Centre of Excellence for Public Health, Queen's University Belfast, Belfast, United Kingdom
[2]Centre for Population Health (CePH), Department of Social and Preventive Medicine, University of Malaya, Kuala Lumpur, Malaysia
[3]Department of Community Medicine, Melaka-Manipal Medical College, Manipal Academy of Higher Education (MAHE), Melaka, Malaysia
[4]Facultas Public Health, University Airlangga, Surabaya, Indonesia
[5]Ministry of Health Malaysia, Putrajaya, Malaysia
[6]National Cancer Society Malaysia, Kuala Lumpur, Malaysia
[7]National Cancer Registry, Cork, Ireland
[8]South East Asia Community Observatory (SEACO), Jeffrey Cheah School of Medicine and Health Sciences, Monash University Malaysia, Kuala Lumpur, Malaysia

**Acknowledgements** We would like to acknowledge all involved in the development of the Be Cancer Alert Campaign materials and would like to acknowledge that the Be Cancer Alert Campaign materials were adapted from materials produced by the Public Health Agency, Northern Ireland for the Be Cancer Aware Campaign. We would also like to thank Associate Professor Siew Yim Loh who assisted in the training of data collectors and the distribution of campaign materials. Furthermore, we thank the Department of Statistics Malaysia for providing the randomly selected households. We would like to acknowledge Datin Dr Premila A/P Kanagasabai (Selayang Hospital)Dr Yun Sii Ing (Sg Buloh hospital), Dr Farah Nadrah Mohd Nasir (Sg Buloh Hospital), Dr Hazwan Amzar (Sungai Buloh), Dr Ainur Syahidza Bt Abd Manan (NCD Unit, PKD Petaling), Dr Nurhaidi Bt Abdullah (KK Kota Damansara), Dr Nur Es Naini (KK Kelana Jaya), Dr Ee Li Min (KK Taman Medan) and Dr Tengku Nor Shafadilah Bt Tg Mohd Nordin (KK Paya Jaras) for providing data on breast cancer screening and all participants for completing the surveys.

**Contributors** MDo and TTS conceptualised and planned the project and are the Co-PIs of the successful grant award from UK MRC-Newton Ungku Omar Fund. CD conducted the power calculation and provided guidance on the study design. DS, MNNH, DP, TTS and MDa planned and coordinated the study and data collection. SS guided the BCAC campaign design and implementation and NSBIT guided the collection of health service data. MNNH collected data from hospitals and clinics. DS and CC conducted the statistical analysis. DS drafted the manuscript. MDo led

the editing and refinement of the manuscript. All authors contributed to, reviewed and approved the final manuscript.

**Funding** This study is funded by UK MRC-Newton Ungku Omar Funding (MR/P013910/1). The collaborative grant application was subjected to peer-review by individual academic reviewers and the final decision about funding was made by an expert panel. The funder had no role in the design of the study, collection, analysis and interpretation of data or in writing the manuscript.

**Competing interests** None declared.

**Patient consent for publication** Not required.

**Ethics approval** Ethics approval for the study was granted by the Medical Research Ethics Committee, University Malaya Medical Centre (ID: 2016126-4668) and by the National Medical Research Register (NMRR-18-1961-42562). Informed consent to participate was provided by all study participants. The study was performed in accordance with the Declaration of Helsinki.

**Provenance and peer review** Not commissioned; externally peer reviewed.

**Data availability statement** Data are available upon reasonable request.

**ORCID iD**
Désirée Schliemann http://orcid.org/0000-0002-8746-3002

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
