## [Reviewer comments · BMJ Open]

ARTICLE DETAILS

TITLE (PROVISIONAL)	The impact of a mass media campaign on breast cancer symptoms awareness and screening uptake in Malaysia: findings from a quasi-experimental study
AUTHORS	Schliemann, Desiree; Htay, Mila; Dahlui, Maznah; Paramasivam, Darishiani; Cardwell, Christopher; Ibrahim Tamin, Nor Saleha; Somasundaram, Saunthari; Donnelly, Conan; Su, Tin; Donnelly, Michael

VERSION 1 - REVIEW

REVIEWER	MERYEM YILMAZ SIVAS CUMHURİYETÜNİVERSİTESİ/TURKEY
REVIEW RETURNED	24-Jan-2020

GENERAL COMMENTS	I congratulate the authors of the study for trying a good way to inform women about breast cancer. The reviewer provided a marked copy with additional comments. Please contact the publisher for full details.
---

REVIEWER	Weijia Shi University of Minnesota, USA
REVIEW RETURNED	06-Mar-2020

GENERAL COMMENTS	I'm pleased to have the opportunity to review this manuscript. This is an important topic and I was hooked to read your paper. I strongly agree with the authors that it's important to leverage mass media to educate the public and conduct rigorous evaluations of campaigns, especially in the context of multicultural societies. I've included my comments below, some of which are major recommendations while others are simply minor clarifications. I've organized these comments by manuscript section. Introduction: • I think this section is well organized and aligned to the research topic. It would be helpful to clarify the sentence "Particularly in settings where resources are limited and population-based cancer screening is unavailable, nation-wide education programmes are encouraged as a first step to improve early detection of breast cancer." I would think a high level of awareness may not be translated into more early detection cases if resources are limited or
--

	people have limited access to these medical services. (minor)  • It would be helpful to add some contextual information for the current communication about breast cancer screening (e.g., move some of the discussion upfront). For example, what is the current recommendation about the starting age and frequency of screening, if any? This may help explain why awareness varies across different age groups. (minor) Methods:  • It seems there's an inconsistency about the campaign timeline between the main text and the appendix table 1. In the main text, campaign occurred from September – October but in the appendix table 1, campaign was from April – May. Related to the timeline, it would be helpful if the authors could also specify the time frame for data collection instead of saying "one-to-twelve weeks before-and-after implementation." With such information, readers can better assess the effectiveness of the intervention. (major) Results:  • I think the results section was well written. My only question is that whether there're any substantial differences between people who completed the post-measures versus those who didn't. I saw the authors presented some information about this in Table 1 but it's worth mentioning in the main text. (minor) Discussion:  • The authors said "the high reach and improvement in symptoms awareness suggested that the campaign impacted successfully and achieved its primary objective." But results showed knowledge improvement was significant for only one comparison between recognisers and non-recognisers using the unprompted measures (i.e., a lump or thickening in your breast). Although there's more evidence that recognisers improved their awareness and knowledge (prompted) compared with non-recognisers, this may be influenced by the sequence of questions (i.e., if aided recall was assessed before prompted awareness, then people may increase knowledge because of the survey exposure rather than campaign exposure). Also, as mentioned in the manuscript, there were other BC awareness activities going along with the BCAC campaign. The effects of knowledge/awareness improvement may be because of exposure to such activities or other factors (e.g., BC history, personal conversation about BC). With that being said, I think the authors need to be more careful about concluding that the campaign impact was successful. Some of the aforementioned points need to be addressed as limitations. (major)
--	---

VERSION 1 – AUTHOR RESPONSE

2. Reviewer 2

Introduction:

I think this section is well organized and aligned to the research topic. It would be helpful to clarify the sentence "Particularly in settings where resources are limited and population-based cancer screening is unavailable, nation-wide education programmes are encouraged as a first step to improve early detection of breast cancer." I would think a high level of awareness may not be translated into more early detection cases if resources are limited or people have limited access to these medical services. (minor)

-We have rewritten the sentence to be more specific.

3. It would be helpful to add some contextual information for the current communication about breast cancer screening (e.g., move some of the discussion upfront). For example, what is the current recommendation about the starting age and frequency of screening, if any? This may help explain why awareness varies across different age groups. (minor)

-We have added this to the introduction.

4. Methods:

It seems there's an inconsistency about the campaign timeline between the main text and the appendix table 1. In the main text, campaign occurred from September – October but in the appendix table 1, campaign was from April – May. Related to the timeline, it would be helpful if the authors could also specify the time frame for data collection instead of saying "one-to-twelve weeks before-and-after implementation." With such information, readers can better assess the effectiveness of the intervention. (major)

-We have corrected the timeline in the appendix and also clarified the data collection timeframe in the methods section.

5. Results:

I think the results section was well written. My only question is that whether there're any substantial differences between people who completed the post-measures versus those who didn't. I saw the authors presented some information about this in Table 1 but it's worth mentioning in the main text. (minor)

-Any significant differences between the pre- and post- study populations have been summarised at the end of 'study population' in the results section.

6. Discussion:

The authors said "the high reach and improvement in symptoms awareness suggested that the campaign impacted successfully and achieved its primary objective." But results showed knowledge improvement was significant for only one comparison between recognisers and non-recognisers using the unprompted measures (i.e. a lump or thickening in your breast). Although there's more evidence that recognisers improved their awareness and knowledge (prompted) compared with non-recognisers, this may be influenced by the sequence of questions (i.e., if aided recall was assessed before prompted awareness, then people may increase knowledge because of the survey exposure rather than campaign exposure). Also, as mentioned in the manuscript, there were other BC awareness activities going along with the BCAC campaign. The effects of knowledge/awareness improvement may be because of exposure to such activities or other factors (e.g., BC history, personal conversation about BC). With that being said, I think the authors need to be more careful about concluding that the campaign impact was successful. Some of the aforementioned points need to be addressed as limitations. (major)

-Thank you, we very much appreciate your comments. We would like to highlight that both, campaign recognisers and non-recognisers were exposed to the same questions, that were asked in the same order. There are still significant differences between prompted awareness in recognisers and non-recognisers. We have rephrased the sentence and emphasised the potential impact it had, and removed the term 'success'. We have included both points under limitations.

VERSION 2 – REVIEW

REVIEWER	Weijia Shi Hubbard School of Journalism and Mass Communication, University of Minnesota, US
REVIEW RETURNED	04-May-2020

GENERAL COMMENTS	Thank you for addressing my concerns/suggestions. I think your research is interesting and useful for practitioners as well as researchers.
---